# Research Progress on Aptamer Electrochemical Biosensors Based on Signal Amplification Strategy

**DOI:** 10.3390/s25144367

**Published:** 2025-07-12

**Authors:** Jiangrong Yang, Yan Zhang

**Affiliations:** 1School of Medical Technology, Dehong Vocational College, Mangshi 678400, China; dhzyyjr@163.com; 2Faculty of Science, Kunming University of Science and Technology, Kunming 650500, China

**Keywords:** aptamer, electrochemistry, biosensor, signal amplification strategy

## Abstract

Aptamers have high specificity and affinity to target analytes, along with good stability and low cost, making them widely used in the detection of target substances, especially in the increasingly popular aptamer-based electrochemical biosensors. Aptamer-based electrochemical biosensors are composed of aptamers as the biorecognition elements and sensors that convert the biological interactions into electrical signals for the quantitative detection of targets. To detect low-abundance target substances, the improvement of the sensitivity of biosensors is a pursuit of researchers. Therefore, different amplification strategies for significantly enhancing the detection sensitivity of biosensors have been explored. Thus, this paper reviews the different amplification strategies with various functional materials to amplify the detection signals. Currently, such strategies commonly use gold nanoparticles to construct electrodes that facilitate the transfer of biological reactions or to obtain enhanced signals through nucleic acid amplification. Some strategies use nucleases for target recycling to further enhance the signals. This review discusses the recent progress in signal amplification methods and their applications, and proposes future directions of study to guide subsequent researchers in overcoming the limitations of previous approaches and to produce reproducible biosensors for clinical applications.

## 1. Introduction

Many electrochemical biosensors for the detection of biomolecules have been reported. The common types of electrochemical biosensors include electrochemical biosensors, electrochemiluminescence biosensors, opto-electrochemical biosensors, and aptasensors [1,2,3,4]. An aptamer is a single-stranded DNA or RNA ligand screened using an exponential enrichment system, which selects nucleotide sequences capable of binding various targets (such as metal ions, organic molecules, proteins, bacteria, cells, etc.) from a random library [3,5,6,7]. Since the first development of RNA aptamers in the early 1990s [8,9], the number of selected aptamers has steadily increased, with the targets spanning small molecules, large molecules, proteins, microorganisms, and cells. Aptamers can fold into a variety of structures, including stem–loop, hairpin, bulge, G-quadruplex, etc., to allow for the specific recognition of target molecules through base stacking, electrostatic interactions between charged groups, van der Waals forces, hydrogen bonds, and spatial complementarity. Compared to traditional antibodies or enzymes, aptamers are less influenced by changes in temperature, pH, and ionic strength. More importantly, the affinities of aptamers to the target molecules are similar to those of antibodies; however, they offer significant advantages, such as more convenient synthesis and modification, high affinity, and good selectivity [10,11,12]. Different aptamers can be flexibly used for different analytes, allowing for the design of diverse sensing strategies. Given their significant advantages over antibodies, aptamer-based electrochemical biosensors have been widely used in biosensor platforms. An aptamer-based electrochemical biosensor uses the biorecognition element to capture the target analytes, inducing changes in the sensor surface parameters, such as the current, potential, impedance, or conductivity. The concentration of the target analyte is then quantitatively determined by monitoring these signal changes [13,14]. Despite the high selectivity, low cost, and rapid response of aptamer-based electrochemical biosensing, detecting low-abundance target biomolecules remains a significant challenge. Therefore, there is an urgent need to develop electrochemical biosensors with high sensitivity and selectivity; in response, aptamer-based electrochemical biosensors with signal amplification strategies have emerged [15,16,17,18]. A signal amplification strategy can achieve the detection of low-abundance substances. The current research focuses on modifying electrodes with materials, such as gold nanoparticles, to improve the electrode performance and achieve a lower limit of detection. Therefore, this paper presents six common materials used in electrochemical aptamers, including carbon nanomaterials, gold nanoparticles, quantum dots, nucleic acids, enzymes, and a combination of these materials, and summarizes their respective advantages and limitations in practical applications. We hope this review will inspire researchers to overcome the limitations of previous methods and design simpler, more practical, and reproducible biosensors for clinical applications. Finally, this paper discusses future directions of aptamer-based electrochemical sensing technology to provide guidelines for researchers aiming to overcome previous limitations and develop simpler, more practical, and reproducible biosensors for clinical applications.

## 2. Nucleic Acid-Based Signal Sensing and Amplification Strategies

### 2.1. Carbon Nanomaterials

Electrochemical aptasensors have many advantages, including high sensitivity, good selectivity, low cost, and simple operation; thus, they are widely used in the recognition and quantification of biomolecules. Aptasensors targeting low-abundance and difficult-to-measure biomolecules have been developed using signal amplification strategies. Currently, nanotechnology and nanomaterials are often used to amplify sensor signals. Due to their high specific surface area, controllable shape, and unique physical and chemical properties, the conductivity of nanomaterials is improved after being modified on an electrode. They are also easy to immobilize with nucleic acid functional components and improve the nucleic acid loading capacity for signal amplification. In addition, enzyme-catalyzed signal amplification and nucleic acid amplification reactions are often used in biosensors for signal amplification. Carbon nanomaterials, including carbon nanotubes (CNTs), graphene (GR), reduced graphene, and oxide (rGO), have been used as matrix supports for immobilizing biorecognition units (such as antibodies and aptamers), owing to their large surface area and excellent mechanical, chemical, and electrical properties [19,20,21,22,23]. Currently, both covalent and non-covalent methods are used for the immobilization of biorecognition elements [24,25]. In covalent coupling, carbon nanomaterials are oxidized to introduce carboxyl groups, which are then linked to the aptamer through amide bond formation [26]. In non-covalent coupling, physical adsorption is often used [27]. In addition, rGO is a new type of two-dimensional carbon material that combines the lamellar structure of graphene with a high specific surface area; the reduction of oxygen-containing functional groups (such as hydroxyl, carbonyl, and carboxyl) further improves the electron transfer rate [28]. Muniandy et al. [29] developed a method for detecting *Salmonella* using a reduced graphene oxide/titanium dioxide (rGO-TiO_2_) nanocomposite. The aptamer was immobilized on the rGO-TiO_2_ nanocomposite. In addition, the rGO-TiO_2_ nanocomposite sensor platform produced an improved signal response compared to rGO or TiO_2_ platforms alone, which contributed to good reproducibility. The aptamer bound to *Salmonella* and formed a physical barrier to inhibit electron transfer, thus reducing the signal detected by DPV. The DPV signal decreased with an increasing *Salmonella* concentration, and the detection limit reached 10 cfu·mL^−1^. Pan et al. [30] constructed a sandwich-type electrochemical biosensing platform using graphene oxide/Prussian blue (GO/PB) as the electrochemical probe and a thorn-like Au@Fe_3_O_4_/MUC1 aptamer as the capture unit for the highly sensitive detection of MCF-7 exosomes in unprocessed serum samples. The GO/PB served to provide electrochemical signals and immobilize the antibodies for identifying and binding the captured exosomes. The excellent conductivity and catalytic performance of the thorn-like Au@Fe_3_O_4_ nanoparticles further enhanced the electron transfer at the electrode interface and amplified the electrochemical signal. Akbarzadeh et al. [31] developed a new, rapid, and label-free aptamer-based electrochemical biosensor for the detection of oxytetracycline (OTC). The sensor was developed based on a newly synthesized nanocomposite consisting of multi-walled carbon nanotubes (MWCNTs), gold nanoparticles (AuNPs), reduced graphene oxide (rGO), and chitosan (CS), and was used to modify a glassy carbon electrode. The resulting electrode had greatly enhanced conductivity, and its binding capacity to the OTC-specific aptamer (via self-assembly) was improved. Electrodes modified with nanocomposite materials (MWCNTs-AuNPs/CS-AuNPs/rGO-AuNPs) were fabricated using a layer-by-layer modification method, which maximized the aptamer stability. The designed aptasensor could effectively measure the content of OTC in milk samples, achieving a detection limit of 30.0 pM.

Recently, carbon nanomaterials have been used as supports for catalytically active metal nanoparticles. Li et al. [32] reported the facile synthesis of gold–graphene hybrid materials using histidine-functionalized graphene quantum dots (His-GQDs) as the connector between the gold and graphene, as well as the semiconductors. The study demonstrated that the use of histidine-functionalized graphene quantum dots as the linker between the gold and graphene and semiconductors reduced the size of the gold nanoparticles. The synthesized Au-His-GQD-G improved the dispersion and formed Schottky heterojunctions at the interfaces. The ultrahigh catalytic activity was obtained as a result of the unique structure. Due to the Au-His-GQD-G and DNA cycle signal amplification, the electrochemical sensor demonstrated high sensitivity and specificity during the electrochemical detection of chlorpyrifos. Also, Qaanei et al. [33] proposed an aptamer-based electrochemical sensor (Figure 1) for detecting *E. coli* O157:H7. They employed a nanoparticle-modified glassy carbon electrode, coated with a reduced graphene oxide–polyvinyl alcohol and gold nanoparticle composite (AuNPs/rGO–PVA/GCE), to increase the surface area and support amplified signal output, which in turn improved the sensor sensitivity. The method has a detection limit of 9.34 CFU mL^−1^.

Carbon nanomaterials-assisted aptasensors have attracted considerable attention due to their excellent physical and electrochemical properties, low cost, and ease of operation. However, variations in the performance of these aptasensors have arisen from difficulties in controlling the chirality, diameter, and degree of aggregation of carbon nanotubes, as well as the impurities and substrate of graphene. Furthermore, the interaction between the aptamers and carbon nanomaterials should be further investigated to fully exploit their signal amplification capabilities [34,35]. In summary, carbon nanomaterial-based biosensors still need improvements in terms of their reproducibility and biocompatibility.

### 2.2. Gold Nanomaterials

Gold nanoparticles (GNPs) possess exceptional characteristics, such as a high surface-to-volume ratio and easy modifiability, making them excellent candidates as matrices for labeling or as electroactive tracers. The excellent biocompatibility, chemical inertness, and fast response of GNPs have been exploited to amplify signals in biorecognition events [36,37,38,39]. GNPs can be used as carriers for aptamer probes [40,41,42]. For example, Wang et al. [43] developed a high-performance electrochemical aptasensor for detecting flufenpyr in fruits and vegetables. They used a 21-peptide as a morphological inducer to synthesize raspberry-shaped gold (RC-Au) nanoprisms. The synthesis process involved the synchronous growth and asymmetric development of the nanoprisms, yielding RC-Au nanoparticles with exposed high-index crystal faces that enhanced the catalytic activity. Subsequently, the RC-Au nanoprisms were conjugated with an aptamer/auxiliary DNA hybrid to construct an electrochemical aptasensor capable of detecting flufenpyr by releasing free auxiliary DNA chains, thereby triggering the target-induced DNA cycle. They found that a flufenpyr molecule could transfer many RC-Au-thiol probes to the surface of the aptasensor, which then induced oxidation and reduction reactions of the probe that led to signal amplification. The method has a limit of detection of 3.7 × 10^−18^ M.

Aflatoxin B1 (AFB1) is one of the most potent and carcinogenic mycotoxins that pose a serious threat to human health. Therefore, the highly sensitive detection of AFB1 is particularly important. Yu et al. [44] constructed an electrochemical aptasensor for AFB1 detection using gold nanoparticles/cobalt–metal–organic framework (AuNPs/Co-MOF) as a modified material combined with an enzyme-free signal amplification strategy. The AuNPs/Co-MOF, with a broad specific surface area and excellent conductivity, could load a large amount of DNA chains and enhance the oxidation–reduction signal of thymidine blue (THi), thereby significantly improving the signal response. Moreover, dual-metal core–shell nanocomposites loaded with thymidine blue (Au@PtNPs) were synthesized as a signal label. The Au@PtNPs have high catalytic activity towards THi and thus could improve the efficiency of the loaded THi and amplify its electrochemical signal. Moreover, the hybrid chain reaction (HCR) signal amplification strategy, used for enzyme-free amplification, ensured sensor stability and triggered a strong electrochemical signal response. Under optimal experimental conditions, the aptasensor has a detection range of 0.001–500 ng/mL and a limit of detection of 1.2 × 10^−2^ pg/mL. Compared with other detection methods, the method is characterized by high sensitivity, simple operation, good stability, and fast response time.

The prostate-specific antigen (PSA) is considered an important biomarker for the diagnosis of prostate cancer. Wang et al. [45] developed an ultra-sensitive ratiometric electrochemical aptasensor for PSA detection. The mechanism of the biosensor is shown in Figure 2. A thiolated hairpin probe 2 is labeled with ferricyanide (Fc-HP2) and then immobilized on a gold electrode (AuE) modified with gold nanoparticles (AuNPs) through Au-S bonds. Upon the introduction of the PSA, the aptamer region of the hairpin probe 1 (HP1) binds to the PSA, while the remaining portion of the HP1 hybridizes with the Fc-HP2 to form a Fc-HP2/HP1/PSA complex with blunt ends that triggers the exonuclease III (Exo III) cleavage process. This process is accompanied by the detachment of Fc from the electrode and the cycling of HP1/PSA. The remaining Fc-HP2 fragment on the electrode surface subsequently hybridizes with methylene blue-labeled DNA (MB-DNA). This interaction leads to the enhancement of the MB signal (IMB) and the weakening of the Fc signal (IFc). Under optimal conditions, the aptasensor demonstrates excellent analytical performance for PSA detection. It has a linear detection range of 100 fg·mL^−1^ to 10 ng·mL^−1^ and a detection limit of 34.7 fg·mL^−1^.

In addition, gold nanoparticles and other metallic nanomaterials, such as copper sulfide nanosheets [46] and zinc oxide nanorods [47,48], among others, have been reported as nanoelectrode materials that facilitate current pathways between bulk electrode materials and reaction mixtures [49]. Su et al. [49] developed an aptamer sensor based on a dual signal detection strategy incorporating gold nanoparticle-modified MoS_2_. The presence of adenosine triphosphate (ATP) and thrombin can induce structural changes in the aptamer sensor, resulting in the ferrocyanide moving closer to or the methylene blue moving away from the electrode surface.

### 2.3. Quantum Dots

Quantum dots (QDs) are colloidal semiconductor nanocrystals with sizes ranging from 1 to 10 nm and are primarily composed of cations from groups II–VI, III–V, and IV–VI. QDs can be classified into three main types: homogeneous structures, core–shell structures, and ternary structures [50]. Due to their excellent electrochemical performance and other advantages, such as their miniaturization, good biocompatibility, controllable morphology, low power consumption, and low cost, QDs have been widely employed as labels for signal amplification in aptamer-based biosensing [51].

Self-assembly between aptamers and quantum dots is a common method for fixing aptamers onto quantum dots. For example, Wang et al. developed a carbon quantum dot-tungsten disulfide nanocomposite (CQDs-WS_2_)-modified glass carbon electrode (CQDs-WS_2_/GCE) and proposed a label-free electrochemical aptamer sensing strategy for the sensitive determination of sulfamethoxazole [52]. Moreover, based on specific interactions, such as biotin–streptavidin interaction [53], aptamers can also be connected to quantum dots through hybridization with complementary sequences modified on nanoparticle surfaces [54]. For example, Zhao et al. [55] designed a simple magnetic electrochemical aptamer sensor for detecting the prostate-specific antigen (PSA) in human serum, using Ag/CdO nanoparticles as the electroactive label. As shown in Figure 3, they first assembled modified Ag/CdO nanoparticles onto the surface of superparamagnetic Fe_3_O_4_/graphene oxide nanosheets (GO/Fe_3_O_4_ NSs) through hydrophobic interactions and π-π stacking interactions between the aptamer and GO nanosheets. As the PSA concentration increased, the high affinity of the aptamer for the PSA caused the Ag/CdO nanoparticles to detach from the GO/F_3_O_4_ nanosheets, thereby altering the electrochemical signal. Ag/CdO has excellent electroactivity and efficient electron transfer, and thus may amplify the detection signal.

Lin et al. [56] constructed a ratiometric electrochemical aptasensor based on graphene quantum dots/[Cu_2.5_(benzotriazole-5-COO)_1.5_(benzotriazole-5-COOH)_0.5_(μ-Cl)_0.5_(μ_3_-OH)-(H_2_O·(GQDs/Cu-MOF)] nanocomposite materials for the detection of *Staphylococcus aureus* (*S. aureus*). In their experiment, single-stranded DNA1 (S1) was immobilized on the surface of a GQDs/Cu-MOF/screen-printed carbon electrode (S1/GQDs/Cu-MOF/SPCE) as a sensing interface. Subsequently, the S1/GQDs/Cu-MOF/SPCE was hybridized with probe DNA–Ferrocene (S2-Fc), and the system generated electrochemical signals due to the Cu-MOF (ICu-MOF) and S2-ferrocene (IS2-FC). However, the electron transfer performance of the DNA at the sensing interface was affected, resulting in a decrease in the ICu-MOF. When *S. aureus* was present, the S2-Fc reacted with the bacteria and detached from the sensing surface, resulting in a gradual decrease in the IS2-FC and an increase in the ICu-MOF. This led to a ratiometric electrochemical aptasensor with high sensitivity (0.97 CFU·mL^−1^), excellent stability, and a wide linear range.

Research on the application of quantum dots in aptamer-based electrochemical detection has progressed rapidly [57,58]. However, despite their many advantages, achieving high selectivity and reproducibility with QD-based aptamer sensors remains challenging for practical samples. Therefore, the pretreatment of actual samples is crucial and should be a focus of future research. Additionally, the selectivity and reproducibility of these aptamer sensors for real samples still require further validation. Nevertheless, these aptamer sensors have shown promising performance in the electrochemical detection of small molecules, and their application has extended to include other targets, such as proteins and DNA.

### 2.4. Enzymes

Enzymes are widely used in electrochemical detection due to their catalytic activity and substrate specificity. Various nuclease enzymes have been employed in signal amplification cycles to improve detection sensitivity. Nucleases [59,60], exonucleases [61,62,63], and deoxyribonucleases [64,65] are three representative nucleases used in aptasensors. For example, Exo III is not active on single-stranded DNA and can recognize a thermostable duplex. At the same time, Exo III is able to selectively degrade the probe DNA from its 3′-end. Based on the properties of Exo III, Zhao et al. [66] used DNA ligase to catalyze the connection of a thrombin-specific aptamer and an extension chain and exonuclease III to catalyze the degradation of probe DNA to achieve the ultra-sensitive detection of thrombin. The electrochemical biosensor has a detection limit of 33 fmol/L. Moreover, this biosensor is capable of distinguishing thrombin in serum samples. Jiang et al. [67] constructed an electrochemical aptasensor based on a DNA tetrahedron (NTH) coupled with gold nanoparticles (AuNPs), combined with enzyme-assisted signal amplification, for the analysis of exosome proteins. For the detection of HepG2 hepatocellular carcinoma exosomes, the aptasensor has a detection limit of as low as 1.66 × 10^4^ particles/mL. Jiao et al. [68] designed an enzyme-assisted signal amplification electrochemical aptasensor for detecting the organophosphorus pesticide pyrethroid. As shown in Figure 4, the designed hairpin probe unfolds upon binding between the pyrethroid and aptamer sequence. After that, the target pyrethroid and the complementary DNA sequence of the hairpin probe (HP) are recycled with the assistance of Vent polymerase and T7 exonuclease (T7 Exo). T7 Exo selectively digests the double-stranded DNA to release Fc-labeled mononucleotides and complementary strands that can hybridize with fresh HPs. Due to the use of T7, the Cy5 fluorescence signal is significantly enhanced. The exonuclease digestion of DNA double strands labeled with ferrocene (Fc) produces single nucleotides, allowing the Fc label to approach the electrode surface, thereby enhancing the electrochemical response. Using this signal amplification strategy, the constructed sensor achieves a detection limit as low as 0.01 ng mL^−1^.

In addition, polymerases consisting of multiple enzyme molecules have also been employed to enhance analytical sensitivity. For example, Xiong et al. [69] developed an α-fetoprotein aptasensor based on horseradish peroxidase–functionalized antibody complexes. The immunocomplex that forms on the electrode surface allows a large amount of the horseradish peroxidase to amplify the electrocatalytic signal. Meanwhile, enzymatic cleavage is employed to further improve the analytical sensitivity. Fan et al. [70] developed a flexible, label-free electrochemical biosensing platform based on the cleavage of split aptamers by exonuclease I. In the presence of the target, the two split aptamers specifically recognize it to form a ternary complex that cannot be digested by exonuclease I, thereby generating a significant impedance signal output. The proposed label-free aptasensor can sensitively and selectively detect thrombin, adenosine triphosphate, and mercury ions, with detection limits of 32 fM, 1.43 pM, and 1.65 pM, respectively. Moreover, artificial mimetic enzymes (e.g., hemoglobin) [71,72] have been explored for biosensing due to their high stability and ease of synthesis and modification compared to natural enzymes. Nuclease-based amplification has several advantages, including ease of design, simplicity of structure, rapid reaction, and compatibility with homogeneous analysis. However, this type of aptasensor is sensitive to sample matrices. This is because impurities in a sample may interfere with the enzymatic reactions during detection.

### 2.5. Nucleic Acids

The common nucleic acid-based signal amplification techniques include rolling circle amplification (RCA) [73], strand displacement amplification (SDA) [74], hybridization chain reaction (HCR) [75], CR, and DNA origami technology [76]. RCA uses a circular DNA template and a specialized polymerase to synthesize long single-stranded DNA or RNA. RCA generates tens to hundreds of tandem repeat sequences complementary to the template. The most common method is to use RCA products as carriers to load a large amount of electrochemical tracers and amplify the electrochemical signal. For example, Feng et al. [73] developed a novel electrochemical aptasensor for detecting low concentrations of ochratoxin A (OTA) by designing a DNA circular probe mediated by OTA to initiate the RCA reaction and using an RCA-driven Ag^+^-DNA enzyme for signal amplification. The limit of detection of OTA was as low as 38 fg/mL. Huang et al. [77] reported an electrochemical and colorimetric aptasensor for the specific detection of gastric cancer exosomes (Figure 5). In their experiment, a gastric cancer exosome-specific aptamer was selected, and various types of exosomes were captured on a gold electrode modified with anti-CD-63 antibody. Among these exosomes, only the gastric cancer exosomes triggered RCA, generating a large number of G-quadruplex units. The product was then incubated with hemoglobin to form a hemoglobin–G-quadruplex structure, which catalyzed a H_2_O_2_ system to produce an electrochemical signal. The linearity and repeatability of the aptasensor were also verified. The aptasensor had high selectivity and sensitivity for gastric cancer exosomes, with a detection limit of 9.54 × 10^2^ mL^−1^.

A hybridization chain reaction (HCR) mimics chain-growth polymerization to produce long double-stranded DNA. Due to its kinetics-controlled reaction, high sensitivity, and selectivity, HCRs have opened an avenue to meet the requirement for efficient signal amplification in biosensors [75]. Niu et al. [75] developed an electrochemical aptasensor based on exonuclease III and a hybridization chain reaction for detecting carcinoembryonic antigen (CEA). The mechanism of the sensor is as follows: the target CEA specifically binds to the aptamer region of hairpin probe 1 (H1), resulting in the triggering of catalytic hairpin assembly by the remaining H1 region, forming a large amount of H1 and hairpin probe 2 (H2) double-stranded complexes (denoted as H1@H2). Subsequently, exonuclease III digests the H1@H2 complex and releases H1 to initiate the first signal amplification. Meanwhile, the generated trigger chains initiate a hybridization chain reaction, resulting in the second signal amplification. The sensor demonstrates excellent analytical performance for CEA detection, with a linear detection range of 10–100 ng/mL and a low detection limit of 0.84 pg/mL. Similar to RCA, HCR products can also serve as carriers for electrochemical tracers. Horseradish peroxidase (HRP) [78], alkaline phosphatase (ALP) [79], artificial mimetic enzymes such as hemoglobin [72], metal nanoparticles with catalytic properties, and electroactive cations [Ru(NH_3_)_6_]^3+^ [80] have been widely used as signal reporter units to achieve signal amplification.

In addition to the use of a single nucleic acid amplification technology, researchers are increasingly favoring the combining of a variety of nucleic acid technologies. For example, Zhang et al. developed an electrochemical aptasensor based on dual signal amplification through strand displacement amplification (SDA) and a hybridization chain reaction (HCR) for the sensitive detection of tumor exosomes [81]. Zeng et al. reported an electrochemical aptasensor for the ultra-sensitive detection of kanamycin (Kana) using dual signal amplification via strand displacement amplification (SDA) and a hybridization chain reaction (HCR) [82].

In short, there are significant advantages of molecular biotechnology, including high efficiency, programmability, biocompatibility, and non-toxicity, but more research is needed to highlight the advantages of molecular biotechnology.

### 2.6. Combined Application of Several Materials

To achieve signal amplification, the combined use of gold nanoparticles, enzymes, nucleic acid technologies, and metal oxides has attracted considerable attention from researchers. This is because each material or method contributes its unique characteristics within the same detection system [1,2,6,7,43,82,83,84]. For example, Du et al. [2] constructed a novel sandwich-type electrochemical aptasensor based on a hemoglobin/G-quadruplex and AuNPs-MoS_2_ dual signal amplification strategy for the highly sensitive and specific detection of thrombin (TB). The thiol-modified thrombin-binding aptamer (TBA-1) was immobilized on the surface of a glassy carbon electrode (AuNPs/GCE) as the capture component via interaction with gold nanoparticles. Another TB-binding aptamer (TBA-2) containing a G-quadruplex structure was also designed to immobilize hemoglobin. The generated hemoglobin/G-quadruplex/TBA-2 sequence was further immobilized on the surface of flower-like MoS_2_ modified with gold nanoparticles (AuNPs-MoS_2_) via a Au-S bond to serve as the signal probe in the sandwich-type electrochemical aptamer detection system. Owing to the dual signal amplification effect of the AuNPs-MoS_2_ and hemoglobin/G-quadruplex in catalyzing the reduction of H_2_O_2_, a highly sensitive TB detection system was successfully constructed. Chai et al. [85] reported an electrochemical aptasensor based on alkaline phosphatase (ALP) and platinum nanoparticles (PtNPs)-functionalized ZnO nanoflowers for thrombin detection. First, ZnO/Pt nanocomposites with a large specific surface area and high catalytic activity were synthesized. Next, a large amount of ALP and TBA II were combined with the ZnO/Pt nanocomposites through a reaction between the Pt nanoparticles and amino groups of the ALP and secondary thrombin aptamer (TBA II) to generate ZnO/Pt/ALP/TBA II biopolymers (TBA II biopolymers). The prepared ZnO/Pt/ALP/TBA II biopolymers have many advantages. First, large amounts of ALP and Pt nanoparticles can be immobilized on ZnO nanomaterials, and the introduction of Pt nanoparticles can improve the conductivity of ZnO nanomaterials. Second, the immobilized ALP can efficiently catalyze the electrochemically inactive substrate 1-naphthyl phosphate (p-NPP), which generates the electron mediator p-naphthol (p-NP). Moreover, the generated p-NP can be further catalyzed by the platinum nanoparticles (PtNPs) on the ZnO surface, thereby amplifying the electrochemical signal. The assembly procedure and the principle of the electrocatalytic detection of the aptasensor are illustrated in Figure 6. The proposed aptasensor has excellent selectivity and sensitivity. Zhu et al. [84] reported an “off-signal” electrochemical aptasensor based on metal–organic framework (MOF) nanocomposites and RecJf exonuclease-assisted target circularization for ultra-sensitive detection of vanillin (VAN). In this experiment, the high catalytic activity of the synthesized gold nanoparticles/iron-based MOF/polydiallyldimethylammonium reduced graphene oxide (AuNPs/NH_2_-MOF-235(Fe)/PEI-rGO), while the AuNPs/Ce-MOF nanocomposites amplified the electrochemical signal. In the presence of RecJf exonuclease, the target cycling triggered by the exonuclease-assisted probe resulted in a significant decrease in the electrochemical signal. Owing to the dual amplification strategy, the performance of the aptasensor was significantly enhanced, thereby allowing for the ultrasensitive detection of VAN. Under optimal conditions, the VAN detection has a linear range of 2 pM–0.2 mM, with a detection limit of 1.14 pM.

In the above examples, each material contributed its specific advantages to amplify the electrochemical signal. Gold nanoparticles and metal oxide nanomaterials offer high biocompatibility, high conductivity, unique electronic and electrocatalytic properties, and sensor activity, making them suitable as carriers for biomolecules or signal amplification. MOFs possess unique properties, such as a large specific surface area and good loading capacity, and can be easily modified with functional groups to facilitate target detection. Polymer nanomaterials can bind biomolecules through covalent bonds, enhancing electrochemical signal transmission and providing a high recognition capacity in bioanalysis. The synergistic combination of these materials with enzymes or DNA can improve the analytical performance of the designed detection method.

## 3. Current Challenges and Future Prospects

In this review, we summarize six commonly used materials, including carbon nanomaterials, gold nanoparticles, quantum dots, nucleic acids, enzymes, and their combinations, employed in nucleic acid signal amplification strategies to improve the sensitivity of electrochemical aptasensors. The use of these materials has greatly improved the performance of aptamer-based electrochemical biosensing. However, several challenges to their practical applications remain to be resolved. Therefore, in this review, we analyze and summarize the advantages and limitations of these materials in practical applications. This review is also intended to serve as a reference for future researchers in this field seeking to overcome the limitations of previous methods and to design simpler, more practical, and more reproducible biosensors for clinical applications.

Despite the significant advances and increasing routine application of electrochemical aptamer-based biosensors, there are still several challenges. The diversity of real sample matrices introduces various interfering components that can affect the binding of a nucleic acid aptamer to the target analytes, thereby reducing the sensitivity of the detection in real samples. Additionally, there is a lack of in-depth and systematic research on the binding sites between nucleic acid aptamers and target molecules. This is because real biological samples are much more complex than laboratory-prepared samples, making it difficult to replicate the stability, sensitivity, and reproducibility of electrochemical aptamer-based sensors obtained under optimal conditions. Moreover, the non-specific binding of aptamers to components other than the target may lead to false-positive results. Thus, sample pretreatment is a necessary step for aptamer sensors; however, pretreatment methods using magnetic beads or magnetic nanoparticles can prolong the detection time or increase the detection costs. In addition, the nanomaterials used in constructing aptamer-based electrochemical sensors should possess a large specific surface area (to provide more binding sites) and good conductivity (to support the loading of more nucleic acid aptamers with good bioactivity); however, their bioactivity is greatly influenced by variations in temperature and pH. Moreover, nanomaterials are prone to degradation or agglomeration, which can greatly impair the detection performance of a sensor. Therefore, problems such as accuracy and stability need to be addressed before aptamer-based electrochemical biosensors can be applied in practical settings. There is no doubt that aptamer-based electrochemical biosensors will make a significant contribution to clinical research once the above challenges are addressed.

## 4. Conclusions

In this review, we summarize six commonly used materials, including carbon nanomaterials, gold nanoparticles, quantum dots, nucleic acids, enzymes, and their combinations, used for nucleic acid signal amplification strategies to improve the sensitivity of electrochemical aptasensors. The application of these materials has greatly improved the performance of aptamer-based electrochemical biosensing. However, several challenges to their practical applications still need to be resolved. Therefore, this review analyzes and summarizes the advantages and limitations of these materials in practical applications. We also propose future development directions, emphasizing that the synthesis of nanoparticles (such as graphene QDs) with good dispersity, uniform particle size, and good functionality should be a key research focus of studies on electrochemical functional nanoparticles. This review is also intended to provide a reference for future researchers seeking to overcome the limitations of previous methods and to design simpler, more practical, and more reproducible biosensors for clinical applications.

## Figures and Tables

**Figure 1 sensors-25-04367-f001:**
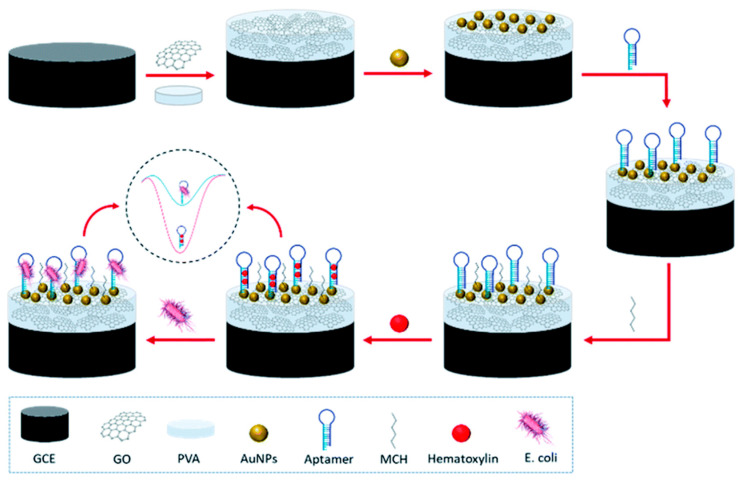
Schematic diagram of the detection mechanism of *E. coli* in an electrochemical aptamer biosensor [33].

**Figure 2 sensors-25-04367-f002:**
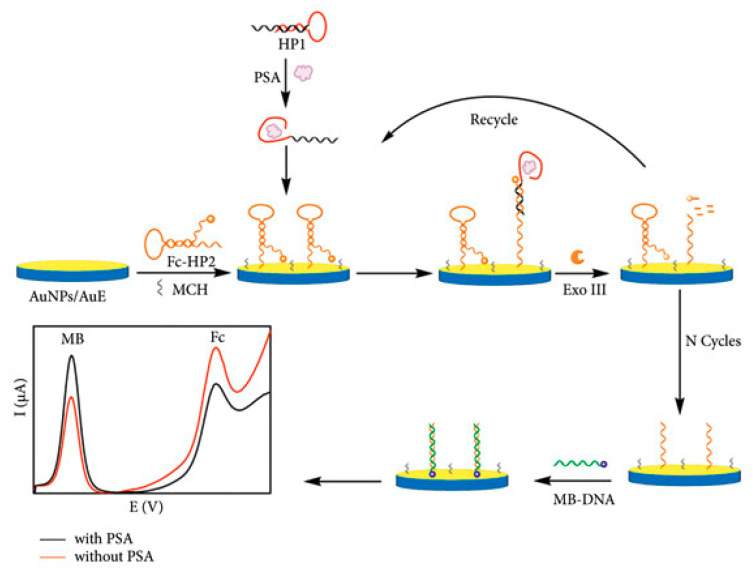
Schematic diagram of the detection mechanism of PSA by an electrochemical aptamer biosensor [45].

**Figure 3 sensors-25-04367-f003:**
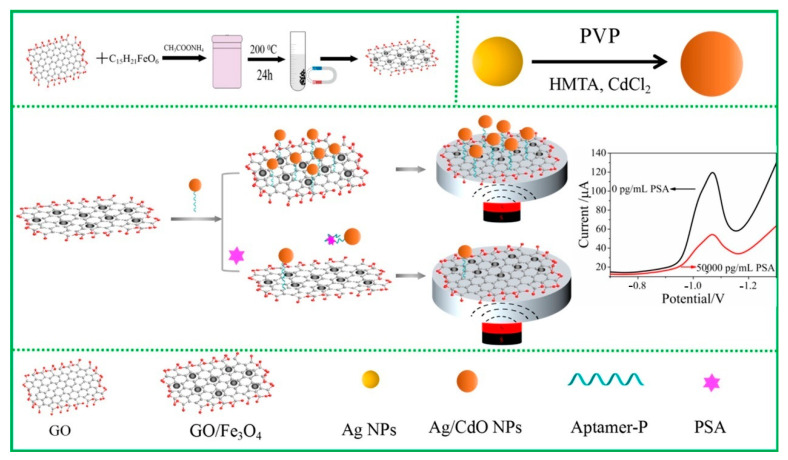
Schematic diagram of electrochemical aptamer sensor constructed with Ag/CdO NP for detecting PSA [55].

**Figure 4 sensors-25-04367-f004:**
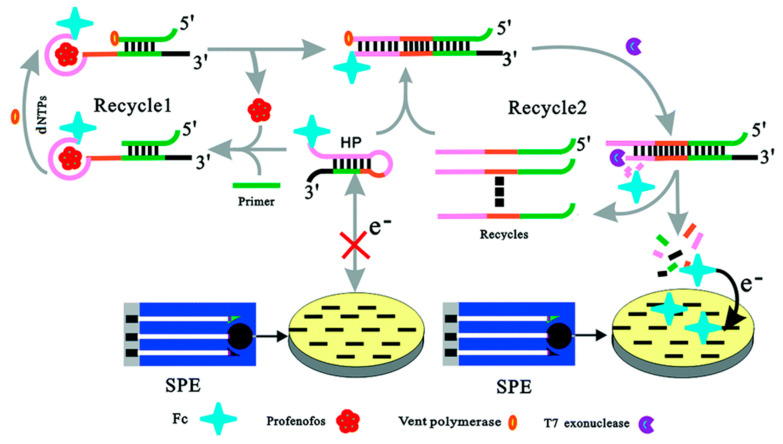
Schematic diagram of electrochemical aptasensor for highly sensitive detection of profenofos based on a target-induced and T7 Exo-assisted recycling amplification strategy [68].

**Figure 5 sensors-25-04367-f005:**
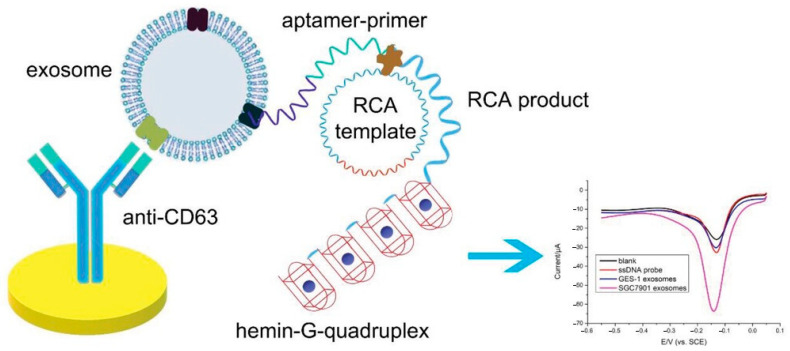
Schematic diagram of electrochemical aptasensor for highly sensitive detection of exosomes [77].

**Figure 6 sensors-25-04367-f006:**
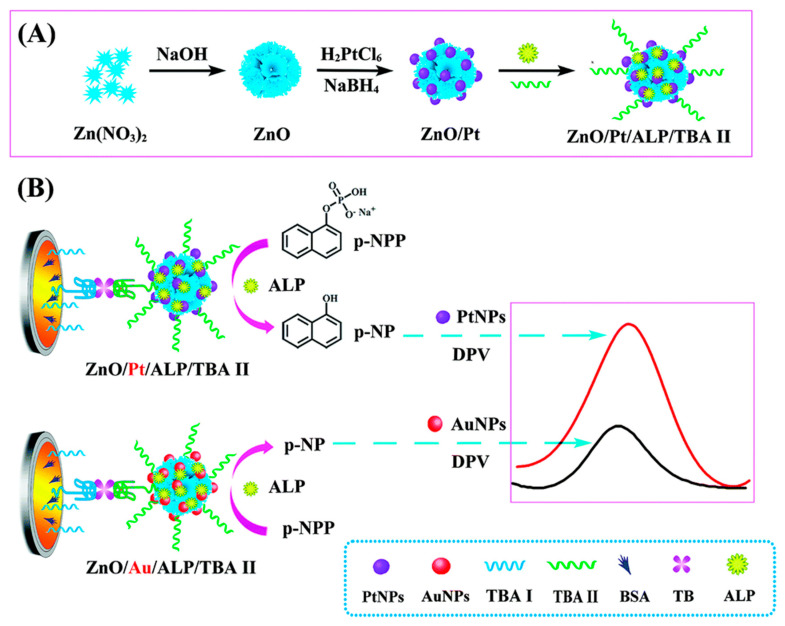
(**A**) Synthesis of ZnO/Pt/ALP/TBA II. (**B**) Signal amplification strategy based on PtNPs [85].

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
