# Peer review of "Research Progress on Aptamer Electrochemical Biosensors Based on Signal Amplification Strategy"

_sensors, 2025, doi:10.3390/s25144367_

Round 1

Reviewer 1 Report

Comments and Suggestions for Authors

Research Progress of Optimal Electrochemical Biosensor Based on Signal Amplification Strategy by Jiangrong Yang and Yan Zhang comprehensively reviews recent advancements in electrochemical biosensors, focusing on signal amplification strategies using various nanomaterials and nucleic acid technologies. The review is well-structured, covering key materials such as carbon nanomaterials, gold nanoparticles, quantum dots, enzymes, nucleic acids, and combinations. The authors highlight the advantages and limitations of each approach, offering valuable insights for future research directions. Given the growing interest in sensitive and selective biosensors for clinical and environmental applications, the article is timely and relevant.

The review thoroughly discusses six major signal amplification strategies, providing a detailed analysis of their mechanisms, applications, and performance metrics. The article cites recent and relevant literature, demonstrating a good grasp of the current state of research in the field.   The manuscript is logically organized, with each section dedicated to a specific strategy, making it easy to follow. The authors summarise existing work and critically evaluate each approach's challenges and limitations, which is valuable for guiding future research. The review emphasizes the potential clinical applications of these biosensors, bridging the gap between laboratory research and real-world use.  While the review covers multiple strategies, a comparative table summarizing the performance metrics (e.g., detection limits, linear ranges, and reproducibility) of different approaches would enhance readability and utility for readers. Some sections, such as those on enzymes and nucleic acid amplification, could benefit from more detailed explanations of the underlying biochemical mechanisms to aid understanding for non-specialists. The review mentions reproducibility issues but does not delve deeply into potential solutions critical for clinical translation.   The manuscript references several Figures 1–6 not included in the provided text. Including these figures would significantly improve the clarity of the described concepts.  Therefore, include a table comparing the detection limits, linear ranges, and other key performance indicators of the biosensors discussed. Provide more detailed explanations of the signal amplification mechanisms, particularly for enzymatic and nucleic acid-based strategies. Address potential strategies for improving reproducibility, such as standardized protocols or quality control measures.   Ensure all referenced figures are included to visualize the described biosensor designs and mechanisms. Elaborate more on specific research gaps and propose concrete steps for overcoming limitations, such as novel material synthesis or integration with portable devices. 

This review is valuable to electrochemical biosensors, offering a detailed and critical overview of signal amplification strategies. With minor revisions to enhance clarity, depth, and comparative analysis, the manuscript will be a more robust resource for researchers. The article is suitable for publication after addressing the suggested improvements. 

Below is a detailed line-by-line breakdown of the weaknesses in the manuscript, along with specific suggestions for improvement:

The abstract is generic and does not highlight the review's specific contributions. Explicitly state the unique aspects of this review, such as the comparative analysis of multiple signal amplification strategies and insights into future research directions. 

Lines 30–35: The advantages of aptamers over antibodies are well-summarized, but the discussion lacks quantitative comparisons such as binding affinities, stability under varying conditions. To strengthen the argument, include a table or specific data points comparing aptamers and antibodies. 

Lines 45–50: The transition to signal amplification strategies is abrupt.   Add a sentence or two explaining why signal amplification is critical for low-abundance targets, linking it to the limitations of current aptasensors. 

Lines 70–75: Covalent vs. non-covalent immobilization discussion lacks depth. Provide examples of specific reactions such as EDC/NHS chemistry for carboxyl-amine coupling also challenges such as steric hindrance. 

Lines 85–90: The examples such as rGO-TiOâ‚‚ nanocomposite are described without critically analysing their limitations such as reproducibility, cost. Add a sentence about the practical challenges of synthesizing these composites. 

Lines 95–100: The conclusion about reproducibility and biocompatibility is vague. Specify which carbon nanomaterials face these issues such as graphene oxide vs. CNTs and propose potential solutions such as surface functionalization. 

Lines 110–115: The description of raspberry-shaped gold nanoprisms is unclear. Include a schematic or TEM image to illustrate the morphology and its impact on catalytic activity. 

Lines 130–135: The AFB1 aptasensor example is detailed but lacks comparison to other methods such as ELISA. Add a sentence comparing the detection limit of this aptasensor to conventional techniques. 

Lines 140–145: Non-specialists do not understand the Exo III cleavage process. Add a brief mechanistic diagram to illustrate the cycling process. 

Lines 160–165: The classification of QDs such as homogeneous, core-shell and ternary is mentioned without explaining their relevance to biosensing. Briefly describe how each type impacts signal amplification such as core-shell QDs for reduced toxicity. 

Lines 180–185: The Ag/CdO NP example is complex and challenging to follow. Simplify the explanation or break it into steps such as Step 1: Aptamer binds PSAStep 2: Ag/CdO detaches → Step 3: Signal change occurs

Lines 190–195: The selectivity and reproducibility challenges are noted but not explored. Discuss specific reproducibility factors such as QD batch variability and potential mitigation strategies. 

Lines 210–215: The nuclease examples of Exo III, T7 Exo are listed without explaining their unique advantages. Compare their performance of Exo III for blunt-end cleavage vs T7 Exo for processivity. 

Lines 230–235: The HRP-antibody complex example lacks context. Clarify why HRP is preferred over other enzymes such as high turnover number and its limitations such as instability. 

Lines 240–245: The interference from sample matrices is mentioned but not quantified. Provide examples such as serum proteins inhibiting Exo III and workarounds include sample dilution. 

Lines 260–265: RCA and HCR are introduced without contrasting their pros/cons. Add a bullet-point list comparing RCA (high yield but expensive) vs. HCR (isothermal but slower). 

Lines 280–285: The CEA aptasensor example is overly technical. Use subheadings such as Catalytic Hairpin Assembly and Exo III Digestion to improve readability. 

Lines 290–295: RCA's cost and time constraints are highlighted but not quantified. Include approximate costs such as $X per circular template) or synthesis times. 

Lines 310–315: The dual-signal amplification example (AuNPs-MoSâ‚‚) is compelling but lacks a critical flaw discussion. Address potential issues such as MoSâ‚‚ oxidation degrading performance. 

Lines 330–335: The vanillin aptasensor’s off-signal mechanism is confusing.   Replace jargon such as target circularization with simpler terms such as target-triggered signal decrease

Lines 370–375: The sample matrix interference discussion is repetitive. Consolidate with earlier sections or provide a dedicated subsection on Real-Sample Challenges.

Lines 390–395: The nanoparticle synthesis focus is too broad. Specify which nanomaterials need improvement such as graphene QDs for batch-to-batch consistency

Author Response

July 9, 2025

Dear editor:

On behalf of my co-authors, we thank you very much for allowing us to revise our manuscript, we appreciate you very much for your positive and constructive comments and suggestions on our manuscript (ID sensors-3739322). These comments were all valuable and very helpful for revising and improving our paper. Our research team accepted all the suggestions and carefully studied your comments, and we have revised the manuscript to address each of the issues you raised and re-uploaded it to your journal. All our revisions are listed as follows. We hope that our revision can make the manuscript more in line with the requirements of your journal.

We would like to express our great appreciation to you for your comments on our paper. If you still have questions about our article, please contact us. Looking forward to hearing from you.

Thank you and best regards.

Yours sincerely,

Yan Zhang

Faculty of Science, Kunming University of Science and Technology

Jingming South Road 727, Kunming, 650500, China

Email address: zhangyan_ecnu@163.com 

Responds to your comments:

The abstract is generic and does not highlight the review's specific contributions. Explicitly state the unique aspects of this review, such as the comparative analysis of multiple signal amplification strategies and insights into future research directions. 

Response: Thank you very much for your kindly reminding. We have accepted these advices and modified the abstract. 

Lines 30–35: The advantages of aptamers over antibodies are well-summarized, but the discussion lacks quantitative comparisons such as binding affinities, stability under varying conditions. To strengthen the argument, include a table or specific data points comparing aptamers and antibodies. 

Response: Thank you very much for your kindly reminding. The advantages of the adapter have been clearly explained here, and corresponding references have also been provided. If listed table here, it's not very good. Because the advantages of antibodies cannot be compared to aptamers.

Lines 45–50: The transition to signal amplification strategies is abrupt. Add a sentence or two explaining why signal amplification is critical for low-abundance targets, linking it to the limitations of current aptasensors. 

Response: Thank you very much for your kindly reminding. The sentence “The signal amplification strategy can achieve the detection of low abundance sub-stances.” was added in here.

Lines 70–75: Covalent vs. non-covalent immobilization discussion lacks depth. Provide examples of specific reactions such as EDC/NHS chemistry for carboxyl-amine coupling also challenges such as steric hindrance. 

Response: Thank you very much for your kindly reminding. In this part“Currently, both covalent and non-covalent methods are used for the immobilization of biorecognition elements [24, 25]. In covalent coupling, carbon nanomaterials are oxidized to introduce carboxyl groups, which are then linked to the aptamer through amide bond formation [26]. In non-covalent coupling, physical adsorption is often used [27].”of the section, the sentence of“carbon nanomaterials are oxidized to introduce carboxyl groups, which are then linked to the aptamer through amide bond formation”is explained here.

Lines 85–90: The examples such as rGO-TiOâ‚‚ nanocomposite are described without critically analysing their limitations such as reproducibility, cost. Add a sentence about the practical challenges of synthesizing these composites. 

Response: Thank you very much for your kindly reminding. The sentence of “The rGO-TiO2 nanocomposite sensor platform produced an improved signal response compared to rGO or TiO2 platforms alone, which contributed good reproducibility.” was added in manuscript.

Lines 95–100: The conclusion about reproducibility and biocompatibility is vague. Specify which carbon nanomaterials face these issues such as graphene oxide vs. CNTs and propose potential solutions such as surface functionalization. 

Response: Thank you very much for your kindly reminding.

There is no discussion on this issue here.

Lines 110–115: The description of raspberry-shaped gold nanoprisms is unclear. Include a schematic or TEM image to illustrate the morphology and its impact on catalytic activity. 

Response: Thank you very much for your kindly reminding. The modifications and additions are as follows.

The study demonstrated that the use of histidine-functional graphene quantum dot as linker between gold and graphene and semiconductor reduces the size of gold nanoparticles. The synthesized Au-His-GQD-G improves the dispersion and forms the Schottky heterojunctions at the interfaces. The ultrahigh catalytic activity was obtained by the unique structure.

Lines 130–135: The AFB1 aptasensor example is detailed but lacks comparison to other methods such as ELISA. Add a sentence comparing the detection limit of this aptasensor to conventional techniques. 

Response: Thank you very much for your kindly reminding. The modifications and additions are as follows.

Compared with other detection methods, the method is characterised by high sensitivity, simple operation, good stability and fast response time. 

Lines 140–145: Non-specialists do not understand the Exo III cleavage process. Add a brief mechanistic diagram to illustrate the cycling process. 

Response: Thank you very much for your kindly reminding. For researchers, this is a fundamental knowledge that does not require explanation.

Lines 160–165: The classification of QDs such as homogeneous, core-shell and ternary is mentioned without explaining their relevance to biosensing. Briefly describe how each type impacts signal amplification such as core-shell QDs for reduced toxicity. 

Response: Thank you very much for your kindly reminding.

Due to their excellent electrochemical performance and other advantages such as miniaturization, good biocompatibility, controllable morphology, low power consumption, and low cost, QDs have been widely employed as labels for signal amplification in aptamer-based biosensing.

These instructions can provide corresponding explanations.

Lines 180–185: The Ag/CdO NP example is complex and challenging to follow. Simplify the explanation or break it into steps such as Step 1: Aptamer binds PSA → Step 2: Ag/CdO detaches → Step 3: Signal change occurs

Response: Thank you very much for your kindly reminding.

The expression of the original text in reference 55: When PSA was present, the high affinity between PSA and aptamers would induce the dissociation of Ag/CdO NPs from GO/Fe3O4 NSs, resulting in the decrease of DPV signals on MGCE. With increasing concentrations of PSA, the DPV signals gradually decreased. There was a linear relationship between the concentration of PSA and DPV currents.

My expression is as follows: As PSA concentration increases, the high affinity of aptamer for PSA causes the Ag/CdO nanoparticles to detach from the GO/F3O4 nanosheets, thereby altering the electrochemical signal. Ag/CdO has excellent electroactivity and efficient electron transfer, and thus may amplify the detection signal.

I don't think it needs to be modified, it has already been expressed clearly.

Lines 190–195: The selectivity and reproducibility challenges are noted but not explored. Discuss specific reproducibility factors such as QD batch variability and potential mitigation strategies. 

Response: Thank you very much for your kindly reminding. The modifications and additions are as follows.

Therefore, pre-treatment of actual samples is crucial and will be a focus of future research.

Lines 210–215: The nuclease examples of Exo III, T7 Exo are listed without explaining their unique advantages. Compare their performance of Exo III for blunt-end cleavage vs T7 Exo for processivity. 

Response: Thank you very much for your kindly reminding. The modifications and additions are as follows.

For example, Exo III is not active on single-stranded DNA and can recognize the thermostable duplex. At the same time, Exo III can be able to selectively degrade the probe DNA from its 3′-end.

T7 Exo selectively digested the double-stranded DNA to release Fc-labelled mononucleotides and complementary strands that could hybridize with fresh HPs.

Lines 230–235: The HRP-antibody complex example lacks context. Clarify why HRP is preferred over other enzymes such as high turnover number and its limitations such as instability. 

Response: Thank you very much for your kindly reminding.

Reference 78 is cited here. In addition, the HRP antibody complex is not the focus here.

Lines 240–245: The interference from sample matrices is mentioned but not quantified. Provide examples such as serum proteins inhibiting Exo III and workarounds include sample dilution. 

Response: Thank you very much for your kindly reminding.

Pre-treatment of actual samples is crucial and will be a focus of future research. However, This is not the focus of the discussion here.

Lines 260–265: RCA and HCR are introduced without contrasting their pros/cons. Add a bullet-point list comparing RCA (high yield but expensive) vs. HCR (isothermal but slower). 

Response: Thank you very much for your kindly reminding.

The manuscript was described as follow.

Despite the significant advantages of molecular biotechnology, including high ef-ficiency, programmability, biocompatibility, and non-toxicity, there are still several challenges. For example, HCR has limitations such as low sensitivity in generating DNA tandem repeats and low catalytic rates, and the preparation of circular templates for RCA is expensive and time-consuming.

If we only list these two methods, it would be too abrupt. The most important thing is that there are few descriptions of other methods, and if only these two methods are listed, it will appear lacking in expression.

Lines 280–285: The CEA aptasensor example is overly technical. Use subheadings such as Catalytic Hairpin Assembly and Exo III Digestion to improve readability. 

Response: Thank you very much for your kindly reminding.

The mechanism of the CEA aptasensor is explained in the manuscript. I think it has been expressed clearly and there are no problems with readability.

Lines 290–295: RCA's cost and time constraints are highlighted but not quantified. Include approximate costs such as $X per circular template) or synthesis times. 

Response: Thank you very much for your kindly reminding.

For example, HCR has limitations such as low sensitivity in generating DNA tandem repeats and low catalytic rates, and the preparation of circular templates for RCA is expensive and time-consuming.

After careful consideration, this paragraph should be deleted.

The modifications and additions are as follows.

In short, the significant advantages of molecular biotechnology, including high ef-ficiency, programmability, biocompatibility, and non-toxicity, but researchers are needed to highlight the advantages of molecular biotechnology.

Lines 310–315: The dual-signal amplification example (AuNPs-MoSâ‚‚) is compelling but lacks a critical flaw discussion. Address potential issues such as MoSâ‚‚ oxidation degrading performance. 

Response: Thank you very much for your kindly reminding.

There is no problem with the expression in the article. If modified according to the review comments, it is not in line with the main idea of the paper.

Lines 330–335: The vanillin aptasensor’s off-signal mechanism is confusing.   Replace jargon such as target circularization with simpler terms such as target-triggered signal decrease

Response: Thank you very much for your kindly reminding. The modifications are as follows.

Zhu et al. reported an “off-signal” electrochemical aptasensor based on metal-organic framework (MOF) nanocomposites and RecJf exonuclease-assisted target circularization for ultra-sensitive detection of vanillin (VAN). In this experiment, the high catalytic activity of the synthesized gold nanoparticles/iron-based MOF/polydiallyldimethylammonium reduces graphene oxide (AuNPs/NH2-MOF-235(Fe)/PEI-rGO), while AuNPs/Ce-MOF nanocomposites amplify the electrochemical signal. In the presence of RecJf exonuclease, target cycling triggered by the exonuclease-assisted probe resulted in a significant decrease in electrochemical signal. Owing to the dual-amplification strategy, the performance of the aptasensor was significantly enhanced, thereby allowing for the ultrasensitive detection of VAN. Under optimal conditions, VAN detection has a linear range of 2 pM - 0.2 mM, with a detection limit of 1.14 pM [84].

Lines 370–375: The sample matrix interference discussion is repetitive. Consolidate with earlier sections or provide a dedicated subsection on Real-Sample Challenges.

Response: Thank you very much for your kindly reminding.

The section of “3. Current Challenges and Future Prospects” discussed the sample matrix interference. I think that there is no problem in here.

Lines 390–395: The nanoparticle synthesis focus is too broad. Specify which nanomaterials need improvement such as graphene QDs for batch-to-batch consistency

Response: Thank you very much for your kindly reminding. We has modified in the manuscript.

Reviewer 2 Report

Comments and Suggestions for Authors

In this manuscript ‘Research Progress of Aptamer Electrochemical Biosensor Based on Signal Amplification Strategy’, the author introduces the development of electrochemical aptasensors using materials such as carbon nanomaterials, gold nanoparticles, quantum dots, nucleic acids, enzymes, and their combinations. The analysis summarizes the advantages and disadvantages of these materials in practical applications and proposes future directions for development. Overall, this manuscript is worth reading. The work seems quite interesting and useful for readers in relative fields.
Specific comments:
1. In the abstract, ‘various methods for significantly enhancing the detection sensitivity of biosensors’ was mentioned. The authors should consider presenting the sensitivity of different biosensors visually through tables or other means to facilitate reader reference.

  1. In Chapter 2, the author aims to demonstrate nucleic acid modification strategies for six types of materials: carbon nanomaterials, gold nanoparticles, quantum dots, nucleic acids, enzymes, and their combinations. Are the titles of Chapter 2 and Section 2.1 reversed?
  2. In Chapter 3, the author mentioned that ‘future development directions require more explicit requirements regarding the use of these materials, and the synthesis of nanoparticles with good dispersibility, uniform particle size, and good functionality should be a key research focus in electrochemical functional nanoparticles.’ When analyzing specific research literature in the manuscript, did the author provide detailed explanations from the aspects of dispersibility, particle size, and functionality? Please provide examples.
  3. Taking Chapter 2.6 as an example, the author mentioned 'large specific surface area,' which is a quantifiable parameter. For such parameters that can be quantified, it is recommended that the author provide the original numerical values.

Author Response

July 9, 2025

Dear editor:

On behalf of my co-authors, we thank you very much for allowing us to revise our manuscript, we appreciate you very much for your positive and constructive comments and suggestions on our manuscript (ID sensors-3739322). These comments were all valuable and very helpful for revising and improving our paper. Our research team accepted all the suggestions and carefully studied your comments, and we have revised the manuscript to address each of the issues you raised and re-uploaded it to your journal. All our revisions are listed as follows. We hope that our revision can make the manuscript more in line with the requirements of your journal.

We would like to express our great appreciation to you for your comments on our paper. If you still have questions about our article, please contact us. Looking forward to hearing from you.

Thank you and best regards.

Yours sincerely,

Yan Zhang

Faculty of Science, Kunming University of Science and Technology

Jingming South Road 727, Kunming, 650500, China

Email address: zhangyan_ecnu@163.com 

Specific comments:

  1. In the abstract, ‘various methods for significantly enhancing the detection sensitivity of biosensors’ was mentioned. The authors should consider presenting the sensitivity of different biosensors visually through tables or other means to facilitate reader reference.

Response: Thank you very much for your kindly reminding.

The abstract was modified as follow.

Aptamers have high specificity and affinity to target analytes, along with good stability and low cost, making them widely used in the detection of target substances, especially in the increasingly popular aptamer-based electrochemical biosensors. Aptamer-based electro-chemical biosensors are composed of aptamers as biorecognition elements and sensors that convert biological interactions into electrical signals for the quantitative detection of targets. To detect low-abundance target substances, improvement of the sensitivity of a biosensor is pursuit by researchers. Therefore, different amplification strategies with significant en-hancing the detection sensitivity of biosensors have been explored. Thus, this paper reviews different amplification strategies with various functional materials to amplify the detection signal. Currently, such strategies commonly use gold nanoparticles to construct electrodes that facilitate the transfer of biological reactions or to obtain enhanced signals through nucleic acid amplification. Some strategies use nucleases for target recycling to further enhance the signal. This review discusses recent progress in signal amplification methods and their ap-plications, and proposes future directions to guide subsequent researchers in overcoming the limitations of previous approaches, and reproducible biosensors for clinical applications.

In addition, the reviewer's comments are very good. However, in each section of the discussion, corresponding results were also provided for sensitivity. In fact, a list can also be omitted.

  1. In Chapter 2, the author aims to demonstrate nucleic acid modification strategies for six types of materials: carbon nanomaterials, gold nanoparticles, quantum dots, nucleic acids, enzymes, and their combinations. Are the titles of Chapter 2 and Section 2.1 reversed?

Response: Thank you very much for your kindly reminding. 

We have modified as following reviewer's comments.

  1. In Chapter 3, the author mentioned that ‘future development directions require more explicit requirements regarding the use of these materials, and the synthesis of nanoparticles with good dispersibility, uniform particle size, and good functionality should be a key research focus in electrochemical functional nanoparticles.’ When analyzing specific research literature in the manuscript, did the author provide detailed explanations from the aspects of dispersibility, particle size, and functionality? Please provide examples.

Response: Thank you very much for your kindly reminding. 

After careful consideration, we believe it would be better to delete this sentence (Future development directions call for more explicit requirements regarding the use of these materials, and the synthesis of nanoparticles with good dispersity, uniform particle size, and good functionality should be a key research focus in electrochemical functional nanoparticles. ). Meanwhile, there are no better examples to illustrate it.

  1. Taking Chapter 2.6 as an example, the author mentioned 'large specific surface area,' which is a quantifiable parameter. For such parameters that can be quantified, it is recommended that the author provide the original numerical values.

Response: Thank you very much for your kindly reminding. 

“First, large amounts of ALP and Pt nanoparticles can be immobilized on ZnO nanomaterials with a high specific surface area.” was modified as “First, large amounts of ALP and Pt nanoparticles can be immobilized on ZnO nanomaterials”. The reason is that the original paper did not provide a specific surface area.
